# Stochastic Online Linear Regression: the Forward Algorithm to Replace Ridge

**Reda Ouhamma**
Univ. Lille, CNRS, Inria, Centrale Lille, UMR 9189 - CRIStAL, F-59000
`reda.ouhamma@univ-lille.fr`

**Odalric. Maillard**
Univ. Lille, CNRS, Inria, Centrale Lille, UMR 9189 - CRIStAL, F-59000

**Vianney. Perchet**
Criteo, ENSAE, ENS PARIS-SACLAY

## Abstract

We consider the problem of online linear regression in the stochastic setting. We derive high probability regret bounds for online *ridge* regression and the *forward* algorithm. This enables us to compare online regression algorithms more accurately and eliminate assumptions of bounded observations and predictions. Our study advocates for the use of the forward algorithm in lieu of ridge due to its enhanced bounds and robustness to the regularization parameter. Moreover, we explain how to integrate it in algorithms involving linear function approximation to remove a boundedness assumption without deteriorating theoretical bounds. We showcase this modification in linear bandit settings where it yields improved regret bounds. Last, we provide numerical experiments to illustrate our results and endorse our intuitions.

## 1 Introduction and preliminaries

The *forward regression* algorithm, popularized in [24, 3], shows competitive performance bounds in the challenging setup of online regression with *adversarial bounded* observations. We revisit the analysis of this strategy in the practically relevant alternative situation of *stochastic* linear regression with sub-Gaussian noise, hence possibly *unbounded* observations. When compared to the classical ridge regression strategy - its natural competitor - the existing analysis in the adversarial bounded case suggests the forward algorithm has higher performances. It is then natural to ask whether this conclusion holds for the stochastic setup. However, we show that in the stochastic setup, the existing adversarial analysis does not seem sufficient to draw conclusions, as it does not capture some important phenomena, such as the concentration of the parameter estimate around the regression parameter. It may further lead the practitioner to use an improper tuning of the regularization parameter. In order to overcome these issues, we revisit the analysis of the forward algorithm in the case of unbounded sub-Gaussian linear regression and provide a high probability regret bound on the performance of the forward and ridge regression strategies. Owing to this refined analysis, we show that the forward algorithm is superior in this scenario as well, but for different reasons than what is suggested by the adversarial analysis. We discuss the implications of this result in a practical application: stochastic linear bandits, both from theoretical and experimental perspectives.

**Setup:** In the classical setting of online regression with the square loss, an environment initially chooses a sequence of feature vectors $\{x_t\}_t \in \mathbb{R}^d$ together with corresponding observations $\{y_t\}_t \in$

35th Conference on Neural Information Processing Systems (NeurIPS 2021).

$\mathbb{R}$. Then, at each decision step $t$, the learner receives feature vector $x_t$ and must output a prediction $\hat{y}_t \in \mathbb{R}$. Afterwards, the environment reveals the true label $y_t$ and iteration $t+1$ begins. In this article, we focus on the case when the data generating process is a *stochastic* linear model:

$$\exists \theta_* \in \mathbb{R}^d \text{ such that } \forall t \in \mathbb{N}^* : \quad y_t = x_t^\top \theta_* + \epsilon_t,$$

where $\{\epsilon_t\}_t$ is a noise sequence. At iteration $t$, strategy $\mathcal{A}$ computes a parameter $\theta_{t-1}^\mathcal{A}$ to predict $\hat{y}_t^\mathcal{A} = x_t^\top \theta_{t-1}^\mathcal{A}$. In the sequel, we omit the subscript $\mathcal{A}$ when the algorithm is clear from context. The learner's prediction incurs the loss: $\ell_t^\mathcal{A} \stackrel{\text{def}}{=} \ell(x_t^\top \theta_{t-1}, y_t) = (\hat{y}_t - y_t)^2$, the learner then updates its prediction $\theta_{t-1}$ to $\theta_t$ and so on. The total cumulative loss at horizon $T$ is denoted $L_T^\mathcal{A} = \sum_{t=1}^T \ell_t^\mathcal{A}$. We also let $\ell_t(\theta) = \ell(x_t^\top \theta, y_t)$ (resp. $L_T(\theta) = \sum_{t=1}^T \ell_t(\theta)$) be the instantaneous (resp. cumulative) loss incurred by predicting $\theta$ at time $t$ (resp. $\forall t = 1, \ldots, T$). Online regression algorithms are evaluated using different regret definitions, in the form of a relative cumulative loss to a batch loss; The quantity of interest in this paper is:

$$R_T^\mathcal{A} = L_T^\mathcal{A} - \min_\theta L_T(\theta). \tag{1}$$

From the perspective of online learning theory, online regression algorithms are usually designed for an *adversarial* setting, assuming an arbitrary bounded response variable $|y_t| \leq Y$ at each time step. While the mere existence of algorithms with tight guarantees in this general setting is remarkable, a practitioner may also consider alternative settings, in which analysis for the adversarial setup may be overly conservative. For illustration, we focus on the practical setting of bounded parameter $\|\theta_*\|_2 \leq S$ and i.i.d zero-mean $\sigma$-sub-Gaussian noise sequences:

$$\forall t \geq 1, \gamma \in \mathbb{R} : \quad \mathbb{E}\left[\exp(\gamma \epsilon)\right] \leq \exp(\sigma^2 \gamma^2 / 2).$$

We emphasize that while previous results in literature are valid for the adversarial bounded setting, we will still shed new light on the performance of these strategies in a stochastic unbounded setup, which is neither more general nor more restrictive than the adversarial one, and discuss their implications for the practitioner. Let us recall the two popular online regression algorithms considered.

**Online ridge regression [Algorithm 1]:**  This folklore algorithm is defined in the online setting as the greedy version of batch ridge regression:

$$\theta_t^{\mathbf{r}} \in \arg\min_\theta L_t(\theta) + \lambda\|\theta\|_2^2, \tag{2}$$

where $\lambda$ is a parameter and $\lambda\|\theta\|_2^2$ is a regularization used to penalize model complexity.

---
**Algorithm 1:** Online ridge regression

---
Given $\theta_0 \in \mathbb{R}^d$
**for** $t = 1, \ldots, T$ **do**
    observe $\mathrm{x}_t \in \mathbb{R}^d$ and predict $\hat{y}_t = x_t^\top \theta_{t-1}^{\mathbf{r}} \in \mathbb{R}$
    observe $y_t$ and incur loss $\ell_t \in \mathbb{R}$
    update parameter: $\theta_t^{\mathbf{r}} \in \arg\min_\theta L_t(\theta) + \lambda\|\theta\|_2^2$
**end**

---

A solution to the quadratic optimization problem of Eq. 2 is given in closed form, by $\theta_t^{\mathbf{r}} = G_t(\lambda)^{-1} b_t$, where $G_t(\lambda) = \lambda I + \sum_{q=1}^t x_q x_q^\top$ and $b_t = \sum_{q=1}^t x_q y_q$. We may further denote $G_t$ instead of $G_t(\lambda)$ when $\lambda$ is clear from context.

**The forward algorithm [Algorithm 2]:**  A subtle change to the ridge regression takes advantage of the next feature $x_{t+1}$ to better adapt to the next loss:

$$\theta_t^{\mathbf{f}} \in \arg\min_\theta L_t(\theta) + (x_{t+1}^\top \theta)^2 + \lambda\|\theta\|_2^2. \tag{3}$$

**Algorithm 2:** The forward algorithm

---

Given $\theta_0 \in \mathbb{R}^d$
**for** $t = 1, \ldots, T$ **do**
    observe $\mathrm{x}_t \in \mathbb{R}^d$
    update parameter: $\theta_{t-1}^{\mathsf{f}} \in \arg\min_\theta L_{t-1}(\theta) + (x_t^\top \theta)^2 + \lambda ||\theta||_2^2$
    predict $\hat{y}_t = x_t^\top \theta_{t-1}^{\mathsf{f}} \in \mathbb{R}$
    observe $y_t$ and incur loss $\ell_t \in \mathbb{R}$
**end**

---

Equivalently, the update step can be written: $\theta_t^{\mathsf{f}} = G_{t+1}^{-1} b_t$, where $G_t$ is still defined same as before. Intuitively, the term $(x_{t+1}^\top \theta)^2$ in Eq. 3 is a "predictive loss", a penalty on the parameter $\theta$ in the direction of the new feature vector $x_{t+1}$. This approach can be linked to transductive methods for regression [8, 22]. [22] describe two algorithms for linear prediction in supervised settings, and leverage the knowledge of the next test point to improve the prediction accuracy. However, these algorithms have significant computational complexities and are not adapted to online settings.

**Related work**   Linear regression is perhaps one of the most known algorithms in machine learning, due to is simplicity and explicit solution. In contrast with the *batch* setting (when all observations are provided), *online* linear regression started receiving interest relatively recently. The first theoretical analyses date back to [9, 14, 7, 13]. Under the assumption that the response variable is bounded $|y_t| \leq Y$, it has been shown that the forward algorithm [24, 3] achieves a relative cumulative online error of $dY^2 \log(T)$ compared to the best batch regression strategy. This bound holds *uniformly* over bounded response variables and competitor vectors, and is 4 times better than the corresponding bound derived for online *ridge* regression.

Bartlett et al. [4] studied minimax regret bounds for online regression, and ingeniously removed a dependence on the scale of features in existing bounds by considering the beforehand-known features setting, where all feature points $\{x_t\}_{1 \leq t \leq T}$ are known before the learning starts. Moreover, they derive a "backward algorithm" that is optimal under certain intricate assumptions on observations and features. Later on, [16] were able to prove that under new (tricky) assumptions on observed features and labels the *backward algorithm* is not only optimal but applicable in sequential settings as well. More recently, [11] provided an optimal algorithm in the setting of beforehand known features without imposing stringent conditions as in [4, 16]. They show that the forward algorithm with $\lambda = 0$ yields a first-order *optimal* asymptotic regret bound uniform over bounded observations. However, due to the lack of regularization, their bound (*cf.* Theorem 11 in [11]) may blow up if the design matrix $G_t(0)$ is not full rank. It is hence not uniform over all bounded feature sequences $\{x_t\}_t$.

**Paper outline and contributions:**   In this paper, we continue the line of work initiated on the *forward* algorithm and advocate for its use in the stochastic setting with possibly unbounded response variables, in replacement for the ridge regression (whenever possible). To this end, we consider an online *stochastic* linear regression setup where the noise is assumed to be i.i.d $\sigma$-sub-Gaussian.

In Section 2 we recall the online performance bounds established for ridge regression and the forward algorithm in the *adversarial* case with bounded observations. Next, in subsection 2.3, we discuss some limitations of the adversarial results when comparing regression algorithms in the stochastic setting. For instance, these bounds compare the cumulative loss of a strategy to the value of the batch optimization problem, which may not be indicative of the real performance of the strategy (*cf.* Corollary 2.3.1) and may encourage a sub-optimal tuning of the regularization parameter.

In Section 3, we study the performance of these algorithms using the cumulative regret with respect to the true parameter (*cf.* Eq. 6), which we believe is more practitioner-friendly than comparing to the batch optimization problem. We show in Theorem 3.1 how these two measures of performance are related. We provide in Theorems 3.2 and 3.3 a novel analysis of online regression algorithms without assuming bounded observations. This key result is made possible by considering high probability bounds instead of bounded individual sequences. We show that the regret upper-bound for ridge regression is inversely proportional to the regularization parameter. Consequently, we argue that following these results, forward regression should be used *in lieu* of ridge regression.

In Section 4, we revisit the linear bandit setup previously analyzed assuming *bounded* rewards: we relax this assumption and provide an -optimism in the face of uncertainty- style algorithm with the

forward algorithm instead of ridge, which is especially well-suited for the bandit setup, and provide novel regret analysis in Theorem 4.1. We proceed similarly in Appendix F, revisiting a setup of non-stationary (abruptly changing) linear bandits.

## 2 Adversarial bounds and limitations

In this section, we recall existing results regarding the aforementioned ridge and forward algorithms. We then discuss their limits and benefits when considered from a stochastic perspective.

### 2.1 Adversarial regret bounds (existing results)

One of the first theoretical analyses of online regression dates back to [24] and [3], and is recalled in the theorem below. It is stated in the form of an "online-to-offline conversion" performance bound.

**Theorem 2.1.** *(Theorem 4.6[1] of [3]) The online ridge regression algorithm satisfies:*

$$L_T^r - \min_\theta \left( L_T(\theta) + \lambda \|\theta\|_2^2 \right) \leq 4(Y^r)^2 d \log \left( 1 + \frac{TX^2}{\lambda d} \right),$$

*where* $X = \max_{1 \leq t \leq T} \|x_t\|_2$, *and* $Y^r = \max_{1 \leq t \leq T} \left\{ |y_t|, \left| \boldsymbol{x}_t^\top \boldsymbol{\theta}_{t-1} \right| \right\}$.

The reader should note that this result compares the learner's online cumulative loss to the regularized batch ridge regression loss. As such, it is an online-to-offline conversion regret. This is different from the sequential regret that would compare to the minimum achievable loss. This theorem highlights a dependence on the *range* of predictions of the algorithm, as $Y^r \geq \max_{1 \leq t \leq T} \left| \boldsymbol{x}_t^\top \boldsymbol{\theta}_{t-1} \right|$.

**Remark 1.** *(Small losses) The regret bound for ridge regression can be improved if the learner knows that the loss is small for the best expert, see Orabona et al. [18]. Note however that such techniques require prior knowledge of all the best expert loss* $L_T^*$, *their optimal bound is* $\sim O(\sqrt{L_T^* \log T})$.

The forward algorithm has an enhanced performance in this setup according to this next result.

**Theorem 2.2.** *(Theorem 5.6 of [3]) The forward algorithm satisfies[2]:*

$$L_T^f - \min_\theta \left( L_T(\theta) + \lambda \|\theta\|_2^2 \right) \leq (Y^f)^2 d \log \left( 1 + \frac{TX^2}{\lambda d} \right),$$

*where* $X = \max_{1 \leq t \leq T} \|x_t\|_2$, *and* $Y^f = \max_{1 \leq t \leq T} |y_t|$.

Notice that in this result $Y$ is different than in Theorem 2.1 and is independent from the algorithm's predictions. Moreover, Theorem 2.2 exhibits a bound that is at least 4 times better than Theorem 2.1. More precisely, Theorem 2.1 suggests that, in order to compare the two bounds, prior knowledge of $Y^f$ is required to further clip the predictions of online ridge regression in $[-Y^f, Y^f]$; and that even with such knowledge the forward algorithm may be 4-times better than ridge regression. We believe that this unfortunately led researchers to turn away from analyzing more deeply what may happen.

### 2.2 Limitation in the adversarial setup: rigid regularization

To evaluate online regression strategies, a tight *lower* bound was derived in [11]. The latter studied uniform minimax lower bounds in the setting of beforehand-known features (that is when $(x_t)_{1 \leq t \leq T}$ known in advance), which is very challenging for a lower bound. They show that, the minimax uniform regret bound is controlled as follows.

**Theorem 2.3.** *(Gaillard et al. [11]) For all* $T \geq 8, Y > 0$ *we have:*

$$R_{T,[-Y,Y]}^\star \geq dY^2 (\log(T) - (3 + \log(d)) - \log(\log(d))).$$

*where* $R_{T,[-Y,Y]}^\star \overset{def}{=} \inf_{\mathcal{A}} \sup_{x_1,\dots,T \in [0,1]^d} \sup_{|y_t| \leq Y} \left\{ \sum_{t=1}^T (y_t - \hat{y}_t^{\mathcal{A}})^2 - \inf_{u \in \mathbb{R}^d} \sum_{t=1}^T (y_t - x_t^\top u)^2 \right\}$

---

[1] Note that there are typos in the statement of this theorem in original paper: compare Lemma 4.2 with Theorems 4.6 and 5.6 therein to see this. Reported theorems are accurate.

[2] See footnote 1.

We will use this result to evaluate the optimality of ridge and forward regressions. First, we need to convert Theorems 2.1 and 2.2 to sequential *regret* bounds. Indeed, in their current form, they compare the cumulative loss of the learner to the value of a regularized batch optimization. This next result transforms them, and is a corollary of Theorems 11.7 and 11.8 of Cesa-Bianchi & Lugosi [6].

**Corollary 2.3.1.** *(Of Theorems 11.7 and 11.8 of [6]) For all $T \geq 1, (x_t)_{1\leq t\leq T} \in \mathbb{R}^d, (y_t)_{1\leq t\leq T} \in [-Y, Y]$ such that $\|x_t\|_2 \leq X$,*

$$for \ \mathcal{A} \in \{\mathbf{r}, \mathbf{f}\} \qquad R_T^{\mathcal{A}} \leq c^{\mathcal{A}}(Y^{\mathcal{A}})^2 d \log\left(1 + \frac{TX^2}{\lambda d}\right) + \frac{\lambda (Y^{\mathcal{A}})^2 T}{\lambda_{r_T}(G_T(0))},$$

*where $r_T = \text{rank}(G_T(0))$ and $\lambda_{r_T}$ is its smallest positive eigenvalue, $c^{\mathbf{r}} = 4$ and $c^{\mathbf{f}} = 1$.*

See proof in Appendix A. This bound suggests that to obtain a $\log(T)$ bound, $\lambda$ should not be chosen larger than about $\log(T)/T$, due to the second term, this is the *stringent regularization limitation*.

Choosing $\lambda = 1/T$ yields a first order regret of $2dY^2$ for the forward algorithm and $8dY^2$ for ridge regression (with clipping and prior knowledge of $Y$), which is at best twice the first order term from the lower bound. This suggests the presence of an optimality gap. Strikingly, Gaillard et al. [11] show that a non-regularized version of the forward algorithm achieves the optimal first order of $dY^2$. However, it also suffers from an important weakness: Indeed, the $(Y^{\mathcal{A}})^2/\lambda_{r_T}(G_T(0))$ term in Corollary 2.3.1 is not uniformly bounded over feature sequences, but only on specific "well-behaved" features. In fact, double uniformity over features and observations is still an open question (see [11]).

## 2.3 Limitations in the stochastic setting

Now that we have recalled the main properties of the forward and ridge algorithms in the adversarial setup, we advocate for the need of a complementary analysis of the previous algorithms in the stochastic unbounded setting by unveiling some key limitations.

**Too unconstrained** The existing analysis being for a different setting, it naturally ignores crucial aspects of the stochastic setup. For instance, the quantity $Y$ is uninformative and may be substantial. Let us look at how the term $Y$ appears in the proofs of Azoury & Warmuth [3]. For ridge regression, the penultimate step to prove Theorem 2.1 writes:

$$L_T^{\mathbf{r}} - \min_\theta \left(L_T(\theta) + \lambda\|\theta\|_2^2\right) \leq \sum_{t=1}^T \underbrace{\left(x_t^\top \theta_{t-1} - y_t\right)^2 x_t^\top G_t^{-1} x_t}_{\text{first term}} \leq 4(Y^{\mathbf{r}})^2 \sum_{t=1}^T x_t^\top G_t^{-1} x_t. \quad (4)$$

In an adversarial setting, the "first term" cannot be controlled without assuming bounded predictions $|x_t^\top \theta_{t-1}| \leq Y^{\mathbf{r}}$, and doing so yields a bound $\left(x_t^\top \theta_{t-1} - y_t\right)^2 \leq 4(Y^{\mathbf{r}})^2$. In a stochastic setup however, we expect the term $\left(x_t^\top \theta_{t-1} - y_t\right)^2$ to reduce and stabilize around $\left(x_t^\top \theta_* - y_t\right)^2$, owing to the convergence properties of the estimate towards $\theta_*$.

For the forward algorithm, the final step in the proof of Theorem 2.2 writes:

$$L_T^{\mathbf{f}} - \min_\theta \left(L_T(\theta) + \lambda\|\theta\|_2^2\right) \leq \sum_{t=1}^T \underbrace{y_t^2 x_t^\top G_t^{-1} x_t}_{\text{first term}} - \sum_{t=1}^{T-1} \underbrace{\boldsymbol{x}_{t+1}^\top \boldsymbol{G}_t^{-1} \boldsymbol{x}_{t+1} \left(\boldsymbol{x}_{t+1}^\top \boldsymbol{\theta}_t\right)^2}_{\text{second term}}. \quad (5)$$

Then, the analysis uses that $|y_t| \leq Y^{\mathbf{f}}$ and disregards the negative contribution of the "second term". *Illustrative example:* Let us analyze these terms in an practice: consider $d = 5$, $\theta_* \in \mathbb{R}^5$, we sample 200 features uniformly in $[0, 1]^5$ and Gaussian noises ($\sigma = 0.1$). Fig. 1 displays the instantaneous first regret term of both algorithms (with $\lambda = 1$) and the second regret term of the forward algorithm, averaged over 100 replicates. We remark that the first terms vanish quickly for ridge regression and are quite stable for the forward algorithm. On the other hand, they are essentially cancelled out by the second term.

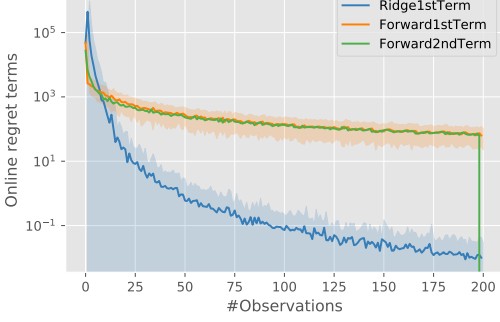

Figure 1: Online regret. $y$-axis is logarithmic.

Overall, the two strategies perform on par on this example. This suggests that Theorems 2.1 and 2.2 can be misleading in this stochastic setup: for ridge regression they introduce a conservative $4(Y^{\mathtt{r}})^2$ bound on $(x_t^\top (\theta_{t-1} - \theta_*))^2$, while in practice we observe that this term decreases rapidly to zero; for the forward algorithm, the bound ignores the effect of a negative term, which, as we see in Fig. 1, is essential to explain why this algorithm may outperform ridge regression.

**Time dependence:** In the stochastic setup, it can be confusing to introduce $Y^{\mathcal{A}}$ because this hides a significant dependence on time. Indeed, for the forward algorithm $Y^{\mathtt{f}} = \max_{1 \leq t \leq T} |x_t^\top \theta_* + \epsilon_t|$. Considering the tractable setting of Gaussian i.i.d noise with variance $\sigma^2$, by classical Sudakov minoration from [20], we deduce that there exists $C > 0$ such that:

$$\forall T \geq 1 : \mathbb{E}[Y^{\mathtt{f}}] \geq \mathbb{E}\left[\max_{1 \leq t \leq T} \epsilon_t\right] - X\|\theta_*\|_2 \geq \sigma C \sqrt{2\log(T)} - X\|\theta_*\|_2.$$

Since $(Y^{\mathtt{f}})^2$ appears in the previous performance bounds, this suggests that $Y^{\mathtt{f}}$ actually increases the order of the regret bound to $\log(T)^2$ in this setting.

By focusing on the *unbounded stochastic* scenario, we hope in this paper to shed novel light on the practical performance of these strategies and better explain these phenomena.

## 3 High probability bounds

In this section, we analyze online ridge regression and the forward algorithm in the *stochastic* setting. We present our results in terms of the following intuitive regret definition:

$$\bar{R}_T^{\mathcal{A}} = L_T^{\mathcal{A}} - L_T(\theta_*). \tag{6}$$

This regret directly compares the cumulative loss of the learner to the cumulative loss of the oracle knowing the true parameter $\theta_*$. This contrasts with the online-to-batch conversion result that compares the loss of the learner to the value of a batch regularized optimization problem. Since we are in a stochastic setup, we further state results in high probability. More precisely, we state Theorems 3.1,3.2, 3.3 below holding with high probability uniformly over all $T$, and not simply for each $T$. As a first step, we prove that for $T$ great enough, we can choose this definition instead of $R_T$ defined in Eq. 1 without altering the bounds.

**Theorem 3.1.** *(Regret equivalence) In the stochastic setting with sub-Gaussian noise, for all $\delta > 0$ with probability at least $1 - \delta$, for all $T > 0$, $(x_t)_{1 \leq t \leq T} \in \mathbb{R}^d$ such that $\|x_t\|_2 \leq X, |G_T(0)| > 0$*

$$L_T(\theta_*) - \min_{\theta \in \mathbb{R}^d} L_T(\theta) = o\left(\log(T)^2\right),$$

*in particular, it comes*

$$R_T^{\mathcal{A}} = \bar{R}_T^{\mathcal{A}} + o\left(\log(T)^2\right)$$

We detail the proof in Appendix B. Theorem 3.1 justifies choosing $\bar{R}_T$ to provide first order guarantees. Indeed, in the following sections we prove high probability upper bounds of order $O(\log(T)^2)$.

### 3.1 Online ridge regression

We start our results by stating a new high probability regret bound for online ridge regression.

**Theorem 3.2.** *In the stochastic setting with sub-Gaussian noise, for all $\delta > 0$ with probability at least $1 - \delta$, for all $T \geq 0$:*

$$\bar{R}_T^r \leq \frac{2d\sigma^2 X^2}{\lambda \log(1 + X^2/\lambda)} \log\left(1 + TX^2/\lambda d\right) \log\left(\frac{(1 + TX^2/\lambda d)^{d/2}}{\delta/2}\right) + o(\log(T)^2),$$

*where $X = \max_{1 \leq t \leq T} \|x_t\|_2$.*

We refer to Appendix C for the proof and for the full regret expression. This result is interesting because the *ranges* of both predictions and observations do not appear, hence predictions clipping and/or a prior knowledge assumption on $Y^{\mathtt{f}}$ are not required. On the other hand, a factor $1/\lambda$ appears in the worst case of a singular design matrix. This seems to be the price for no longer assuming bounded predictions. Another notable improvement is that this bound no longer involves $\lambda\|\theta\|_2^2$ terms. In particular, it is uniform over bounded sequences of observations.

**Remark 2.** *(Regularization in ridge) Note that the bound holds with high probability, uniformly over $T$, and not only for each individual time horizon. In the proof of this result, $1/\lambda$ emerges from bounding $\lambda_{min}(G_t(0))$ in the worst case. When the collected features ensure the design matrix $G_t(0)$ is invertible, $1/\lambda$ virtually disappears. We highlight this experimentally in Section 3.4.*

## 3.2 The forward algorithm

We now derive a high probability regret bound for the forward algorithm using similar techniques.

**Theorem 3.3.** *Assuming sub-Gaussian noise, with probability at least $1 - \delta$, for all $T \geq 0$:*

$$\bar{R}_T^f \leq 2d\sigma^2 \log\left(1 + TX^2/\lambda d\right) \log\left(\frac{(1 + TX^2/\lambda d)^{d/2}}{\delta/2}\right) + o(\log(T)^2),$$

*where $X = \max_{1 \leq t \leq T} \|x_t\|_2$ and the $o(\log(T)^2)$ depends on $\lambda$ (see Appendix D).*

*Proof.* **(sketch)** We refer to Appendix D for the full proof and outline here the main steps leading to this bound. First, the instantaneous regret writes $\bar{r}_t = \ell_t(\theta_{t-1}) - \ell_t(\theta_*) = \left((\theta_{t-1} - \theta_*)^\top x_t\right)^2 + 2\epsilon_t(\theta_{t-1} - \theta_*)^\top x_t$. To control the term involving the noise $\epsilon_t$, we derive ad-hoc self-normalized tail inequalities in the same style of Theorem 1 in [1], which are of independent interest. Such an adaptation is required due to the forward algorithm. We then focus on controlling the first term. We construct confidence intervals for $\theta_*$ and resort to standard technical tools. □

Theorem. 3.3 exhibits a better bound than Theorem. 3.2. In fact, the coefficient of the first order term for the forward algorithm only depends on the dimensionality and the noise variance, whilst for ridge regression, it also depends on the features' scale and on the regularization parameter $\lambda$.

**Remark 3.** *(Unrestrained regularization) Compared to existing results, this analysis lifts the "stringent regularization" that requires $\lambda = 1/T$ or data-dependent regularization (cf. [16]) to obtain uniform bounds. Therefore, Theorems 3.2 and 3.3 are not a mere consequence of bounding $Y^2$ with high probability in previous deterministic theorems. For completeness, we also derive a high probability regret bound for a non-regularized version of the forward algorithm in Appendix E; this algorithm was proven to be asymptotically first order minimax optimal in the adversarial bounded setting [11].*

## 3.3 Tightness of the bounds

Here we clarify the impact of a tighter confidence width for regularized least squares that was proved concurrently with the writing of this paper. First we state the result then we discuss its implications.

**Theorem 3.4.** *(Theorem 1 of Tirinzoni et al. [21]) Let $\delta \in (0, 1), n \geq 3$, and $\widehat{\theta}_t$ be a regularized least-square estimator obtained using $t \in [n]$ samples collected using an arbitrary bandit strategy $\pi := \{\pi_t\}_{t \geq 1}$. Then,*

$$\mathbb{P}\left\{\exists t \in [n] : \left\|\widehat{\theta}_t - \theta_*\right\|_{\bar{V}_t} \geq \sqrt{c_{n,\delta}}\right\} \leq \delta$$

*where $c_{n,\delta}$ is of order $\mathcal{O}(\log(1/\delta) + d \log \log n)$.*

This has important implications for Theorem 3.1, Theorem 3.2 and Theorem 3.3: in short it re-scales their regret upper-bounds from $R_T = O\left((d\sigma)^2 \log(T)^2\right)$ to $R_T = O\left(d\sigma^2 \log(T) \log\log(T)\right)$. The first order $(d\sigma)^2 \log(T)^2$ in our results is the product of **1)** $d \log T$ from the elliptical lemma, for bounding the sum of feature norms and **2)** $\sigma^2 \log(T^d/\delta)$ the confidence ellipsoid width in the estimation of the regression parameter. It is the second term that is altered following the new result from Tirinzoni et al. [21]. These tighter confidence intervals change the upper bounds to $O(d\sigma^2 \log(T) \log\log(T))$. The latter matches the popular lower bounds in excess risk literature (see *e.g.* Theorem 1 in Mourtada [17]) up to sub-logarithmic terms suggesting *the optimality of the forward algorithm in the stochastic setting.*

### 3.4 Experiment

We provide experimental evidence supporting the fact that our novel high probability analysis better reflects the influence of regularization than results its adversarial counterpart.

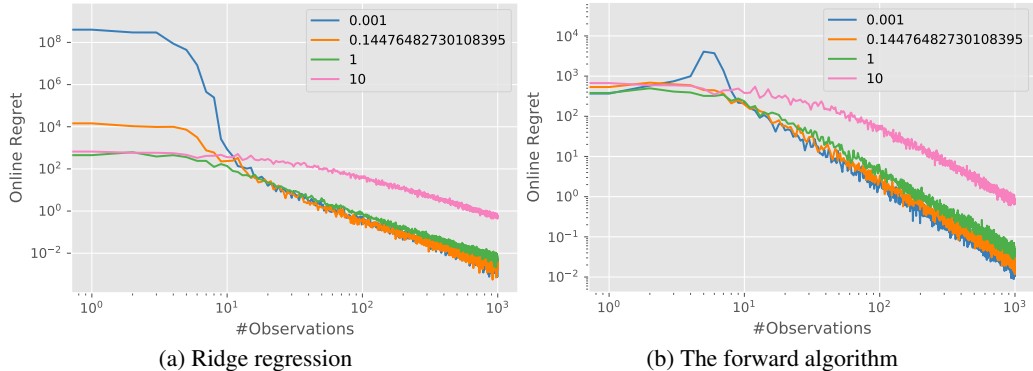

(a) Ridge regression          (b) The forward algorithm

Figure 2: Online regret's (Instantaneous loss difference) dependence on $\lambda$. All axes are logarithmic. Lines are averages over 100 repetitions and shaded areas represent one standard deviation.

In Figures 2a and 2b we observe the effect of regularization on the performance of ridge and forward regressions in a 5-dimensional regression setting, we vary $\lambda \in \{1/T, 1/\log(T), 1, 10\}$, sample a zero mean Gaussian noise with $\sigma = 0.1$ and draw features uniformly from the unit ball. The results clearly highlight the robustness of the forward algorithm to $\lambda$, contrarily to ridge. In particular, for ridge regression, we observe the exact dependence on $\lambda$ described by Theorem 3.2 in the first rounds of learning; as explained in Remark 2, once the collected features are enough for the design matrix $G_t(0)$ to become non-singular, the $1/\lambda$ virtually disappears from the first order regret bound and is replaced by the smallest eigenvalue of $G_t(0)$, making the regret significantly more stable.

## 4 Application: linear bandits

The proposed analysis of forward regression in the stochastic setting suggests that using it could be useful for revisiting several popular setups that include linear function approximation. We apply this change for stochastic linear bandits hereafter and derive the novel regret bound obtained when using forward regression instead of the standard ridge regression.

Consider the setting of *stochastic linear bandits*, where at round $t$ the reward of an action $x_t$ (from the action space $\mathcal{X} \subset \mathbb{R}^d$) is $y_t = \langle x_t, \theta_* \rangle + \epsilon_t$, where $\theta_* \in \mathbb{R}^d$ is an unknown parameter and $\epsilon_t$ is, conditionally on the past, a $\sigma$-sub-Gaussian noise. An upper bound $S$ on the unknown parameter's norm is provided: $\|\theta_*\|_2 \leq S$. The (pseudo) regret in this setting is defined:

$$R_T = \sum_{t=1}^{T} \langle x_t^*, \theta_* \rangle - \sum_{t=1}^{T} \langle x_t, \theta_* \rangle = \sum_{t=1}^{T} \langle x_t^* - x_t, \theta_* \rangle, \tag{7}$$

where $x_t^* = \arg\max_{x \in \mathcal{X}} \langle x, \theta_* \rangle$. Traditionally, the following additional assumption is made.

**Assumption 1.** *for all* $x_t \in \mathcal{X}$    $\langle x_t, \theta_* \rangle \in [-1, 1]$.

The "optimism in the face of uncertainty linear bandit" (OFUL) algorithm was introduced in [1]. OFUL resorts to ridge regression, constructs a confidence ellipsoid for the parameter estimate, and chooses the action that maximizes the upper-confidence bound on the reward. Under Assumption 1, [1] prove that the cumulative regret of OFUL satisfies, for $\delta > 0$ with probability at least $1 -$

$$\delta, \forall T > 0 \quad R_T^{\mathrm{r}} \leq 4\sqrt{Td\log(\lambda + TX^2/d)}\left(\lambda^{1/2}S + \sigma\sqrt{2\log(1/\delta) + d\log(1 + TX^2/(\lambda d))}\right),$$

where $X = \max_{1 \leq t \leq T} \|x_t\|_2$.

**Forward variant [Algorithm 3]:** In a second phase, we propose the variant OFUL$^\mathrm{f}$ in which we replace ridge regression by the forward algorithm. What this means is that the parameter estimate is a function of actions:

$$\theta_t^\mathrm{f}(x) = \arg\min_{\theta \in \mathbb{R}^d} \sum_{s=1}^{t}(y_s - \langle x_s, \theta\rangle)^2 + \lambda\|\theta\|_2^2 + \langle x, \theta\rangle^2.$$

This fits perfectly because the new action can be chosen. Implementation details are in Algorithm 3.

---

**Algorithm 3:** OFUL$^\mathrm{f}$ algorithm

---

Given $\lambda, \delta, S > 0$
**for** $t = 1, \ldots, T$ **do**

$\quad x_t = \arg\max_{x \in \mathcal{X}} \langle x, \theta_t^\mathrm{f}(x)\rangle + \|x\|_{G_{t-1,x}^{-1}}(\sqrt{\lambda} + \|x\|_2)S + \sigma\sqrt{2\log\left(\frac{(1+tX_t^2(x)/\lambda d)^{d/2}}{\delta}\right)},$

$\quad$ where $X_t(x) = \max\{\|x\|_2, \max_{1\le s\le t-1}\|x_s\|_2\}$, $G_{t-1,x} = G_{t-1} + xx^\top$ and
$\quad \theta_t^\mathrm{f}(x) = \arg\min_{\theta\in\mathbb{R}^d}\sum_{s=1}^{t-1}(y_s - \langle x_s, \theta\rangle)^2 + \lambda\|\theta\|_2^2 + \langle x, \theta\rangle^2$
$\quad$ play $x_t$ and observe $y_t$.
**end**

---

Note that OFUL$^\mathrm{f}$ only requires an upper bound $S$ on $\|\theta_*\|_2$. We prove that OFUL$^\mathrm{f}$ enjoys the same regret bound as OFUL and doesn't require Assumption 1. In stark contrast, we cannot show a similar bound for the standard OFUL without said assumption, it actually suffers a $\lambda$-dependent scaling factor in this case.

**Theorem 4.1.** *(Bandits with unbounded rewards) Without Assumption 1, for all $\delta > 0$, OFUL$^r$ achieves with probability at least $1 - \delta$, for all $T \ge 1$,*

$$R_T^r \le 4\sqrt{\frac{\boldsymbol{X}^2}{\boldsymbol{\lambda}\log(1+\boldsymbol{X}^2/\boldsymbol{\lambda})}Td\log(\lambda+TX^2/d)}\left(\lambda^{1/2}S+\sigma\sqrt{2\log(1/\delta)+d\log(1+TX^2/(\lambda d))}\right),$$

*also, we show that for all $\delta > 0$, OFUL$^f$ achieves with probability at least $1 - \delta$, for all $T \ge 1$:*

$$R_T^f \le 4\sqrt{Td\log(\lambda+TX^2/d)}\left((\lambda^{1/2}+X)S + \sigma\sqrt{2\log(1/\delta)+d\log(1+TX^2/(\lambda d))}\right).$$

**Remark 4.** *we can drop the dependence on $X$ and $S$ by bounding the second term in the index of OFUL and OFUL$^f$ (see line 285) by $XS(1 + X/\sqrt{\lambda})$ and then dropping this -constant- term at the expense of a looser index. Therefore, knowing the bounds $x$ and $S$ is not crucial. Furthermore, while we choose to adopt the pseudo-regret definition like in [1], we could also derive similar bounds for the regret involving rewards $y_t = \langle x_t, \theta_*\rangle$ instead of their expected value, $(y_t)_{t\ge 1}$ are unbounded.*

**Experiment** We provide experimental evidence that the OFUL$^\mathrm{f}$ variant improves OFUL for linear bandits; we find that it is generally as good as the standard OFUL$^\mathrm{r}$, and in some cases it can prove to be significantly more robust to aberrant regularization parameters. We consider a 100-dimensional linear bandit with 10 arms, the parameter vector is drawn from the unit ball, actions are such that $\|x_t\| \le 200$. Noise $\epsilon_t \overset{\mathcal{L}}{=} \mathcal{N}(0, 10^{-1})$, $\lambda = 10^{-5}$, $\delta = 10^{-3}$. In Fig. 3, lines are average regret over 100 repetitions and shaded areas cover the region between dashed-lines that are the first and third quartiles.

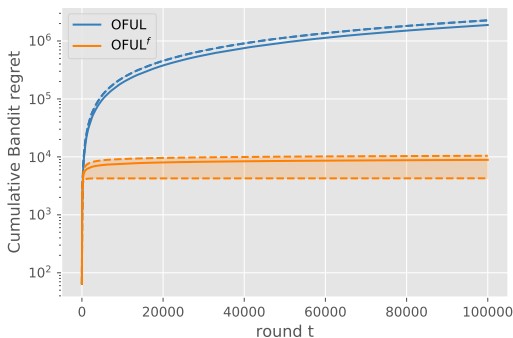

Figure 3: Cumulative regret. $y$-axis is logarithmic.

We observe that -as predicted by Theorem 4.1: OFUL$^\mathrm{f}$ is particularly robust and choosing $\lambda = 1/T$ incurs substantial regret for OFUL. Because of this phenomena, and for the same observations in the online stochastic regression setting, we advocate for the use of the forward algorithm instead of ridge regression whenever possible, to take advantage of its increased robustness to $\lambda$.

**Remark 5.** *Regarding the choice $\lambda = 1/T$: we use this specific regularization for two reasons: 1) to demonstrate the benefits of our stochastic analysis, since previous deterministic bounds suggest this $\lambda$ is best, 2) to showcase the increased robustness of OFUL$^f$ compared to OFUL. In fact, more often than not, OFUL performs as good as OFUL, except when $\lambda$ is small or $X$ is large.*

## 5    Conclusion

We revisited the analysis of online linear regression algorithms in the setup of stochastic, possibly unbounded observations. We proved high probability regret bounds for three popular online regression algorithms (*cf.* Theorems 3.2, 3.3 and E.1). These bounds provide novel understanding of online regression. In particular, Theorem 3.2 seems to be the first regret bound for ridge regression that does not require bounded predictions or prior knowledge of a bound on observations. Our novel bounds seem to correctly capture the nature of dependence with regularization, as indicated by Fig. 2. Moreover, a new results from Tirinzoni et al. [21] can be incorporated in the proof mechanism to bring the high probability upper bounds to $O(d\sigma^2 \log(T) \log \log(T))$, which matches the optimal achievable bounds from the excess risk literature up to sub-logarithmic factors.

Furthermore, we argue that replacing ridge regression by the forward algorithm whenever possible in algorithms that require linear approximations can be beneficial, we depict this in a case study involving linear bandits: First from a theoretical standpoint our results show that the OFUL$^f$ algorithm enjoys the classic first order regret bound while dropping Assumption 1; Second, we find that empirically, implementing OFUL with the forward algorithm makes the algorithm significantly more robust to extreme values of regularization, which is of practical interest.

More broadly, we believe that the improvement resulting from replacing ridge regression with the forward algorithm could be extended to several other settings: For instance, we also provide a similar analysis for non-stationary linear bandits in Appendix F; Graph bandits are of interest as well: they consider linear function approximations using ridge regression, and make Assumption 1, see for example Theorem 1 of [23]; Meta-learning with linear bandits can also be enhanced using forward regression: see for example Lemma 1 and consequent results in [5].

**Societal impact**    While our findings are purely theoretical, they can be taken advantage of for activities with negative societal impact. For instance, bandit algorithms are mainly used nowadays for advertising which can sometimes be linked with invasion of privacy. We insist however that linear regression in its generality is a very valuable technique, its use is ubiquitous is scientific domains, and improving our understanding of it even slightly is beneficial.

## Acknowledgments and Disclosure of Funding

This work has been supported by the French Ministry of Higher Education and Research, Inria, Scool, the French Agence Nationale de la Recherche (ANR) under grant ANR-16-CE40-0002 (the BADASS project), the MEL, the I-Site ULNE regarding project R-PILOTE-19-004-APPRENF. V. Perchet acknowledges support from the French National Research Agency (ANR) under grant number #ANR-19-CE23-0026 as well as the support grant, as well as from the grant "Investissements d'Avenir" (LabEx Ecodec/ANR-11-LABX-0047)". R. Ouhamma also awknowledges support from Ecole polytechnique under the AMX funding.

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
