# OpenReview forum: "Stochastic Online Linear Regression: the Forward Algorithm to Replace Ridge"
_NeurIPS.cc/2021/Conference — Accept (Poster)_

### Official Review · Reviewer_FknY · 2021-07-11

**Rating:** 5
**Confidence:** 4

**Summary:**

This paper proposes a new analysis result on stochastic online ridge regression and its forward version. As an application, the authors propose a new linear bandit algorithm with an improved regret bound (by dropping one common assumption) and show its empirical superiority over existing algorithms.

**Ethical Concerns:**

There are no ethical concerns.

**Limitations And Societal Impact:**

The authors have addressed those adequately.

**Main Review:**

[main comments]

originality: 2/5 -- the novelty I found is the lower bound of order log(T)^2 and the empirical validation that an obvious application of VAW to bandits work well.
quality: 2/5 -- the discussion on existing regret bounds could've been done more carefully and comprehensively (see my comments below)
clarity: 3/5 -- within the scope of the paper, things are reasonably clear, but Theorem 3.2 does not seem complete (see my comment below)
significance: 2/5 -- besides the lower bound, I was not able to confirm its significance yet.

Overall, I think this paper needs some more work on clarifying their claims and put things in the right context. The main issue I have is that "stringent regularization limitation" seems like a madeup problem and the contribution for the linear bandit w.r.t. the theoretical bound is unclear to me, as I describe below. I am willing to raise the score according to the author response.

First, I am not sure about the stringent regularization limitation (L144). The standard analysis of the regret bound of VAW gives us
   Y^2 d log(1 + T) + lambda * \|\theta^*\|^2.
(see Sec 7.10 of Orabona'19 -- see below for the full reference)
Using Y \lessapprox \max_t  (x_t^\top \|\theta^*\|)^2 + log(T) with high probability, the leading term becomes (X\|\theta^*\|) d log(T)^2. Now, this is a factor d better than what was proposed, at the price of \|\theta^*\|^2. Note that the term lambda * \|\theta^*\|^2 is actually in Theorem 3.2 as well -- it is just ignored since it is o(log(T)^2). So, I am really not sure about the actual improvement.

Also, staring at the regret bound of VAW gives us that the best \lambda is something like 1/\|\theta^*\|^2, so it does not have to be dependent on $T$. This is why I think the 'stringent regularization limitation' is a madeup problem. I mean, the dependence on the norm of $\theta^*$ can be considered as stringent regularization limitation, but then the author's new bound also has it, so I don't understand the main point.

Theorem 3.2 is not so convincing.Reading its proof (Section C): I don't see what reasoning was used to conclude that, at L499, "Eq 11 is a of order ~ O(log(T)^{3/2})". theta_{t-1} is clearly a random variable, and with low probability, it can blow up very large. I believe a more proper analysis must be done. Please share with us how the authors will address this.

Sec 2.3: There is an important discussion that is missing -- Ridge regression enjoys a small loss bound, so its actual regret must be smaller than the standard ones imply; see ``Beyond Logarithmic Bounds in Online Learning'' by Orabona, Cesa-Bianchi, and Gentile. One should translate this into the squared regression loss directly (rather than relying on the exp-concavity) to get a tighter bound (see Sec 7.9 of Orabona'19).

Finally, I am not sure if the linear bandit version makes any advancement from prior work. The book LS'20 has an exercise that lifts up the assumption that \langle x_t, \theta_* \rangle \in [-1,1]; see the note between Exercise 19.3 and 19.4. Note that the arm set is fixed, so the learner already knows a bound $X$ s.t. $\max_t \|x_t\|\le X$, so there is not much exciting thing going on about the uniformity over features.

[minor comments and questions]

 - p3:84-86: I am not sure if this is a 'real' problem of [10]. Consider that the algorithm always work within the space spaned by x_t so far (say that is d' dimensional), and the same analysis techniques are invoked for dimension d' < d. Would the same theorem statement not work?
 - p4:119: How is Y here defined? Is this a typo of Y^r or something else that was defined in p3:72?
 - Lemma B.2.: TX^2/\lambda_min(G_{T_0})d => I would clarify it by adding parenthesis regarding (TX^2/\lambda_min(G_{T_0}))d vs TX^2/(\lambda_min(G_{T_0})d)
 - Lemma C.2.: wasn't this proved already in Corollary 8 of ``Online-to-Confidence-Set Conversions and Application to Sparse Stochastic Bandits'' by Abbasi-Yadkori, Pal, Szepesvari? I would add a reference.
 - L217: "no longer involves \lambda \|\theta\|^2" => this is misleading; the bound actually has \lambda \|\theta\|^2 but it is just a lower order term so omitted. There is an actual difference between "not invovling" vs "invoving in a lower order term".
 - Figure 2: it should be online 'Regret / T', because it is decreasing?

[references]

[Orabona'19] Orabona, a modern introduction to online learning, arxiv, 2019.
[LS'20] Lattimore and Szepesvari, bandit algorithms, 2020.

----

(after rebuttal)
I agree that Eq (12) and Lemma C.1 imply that the sum becomes $\log^2(T)$. Thanks for the explanation and I have raised the score from 4 to 5. It is more interesting than I have initially thought.

Overall, there are interesting pieces. However, to appreciate it, I think the paper needs nontrivial amount of modification. It seems there are lots of discussions that should/can be added, which will clarify the authors' contribution. For example, the fact that we do not want to tune \lambda = 1/\|\theta\|^2 is that we want to be parameter-free. I recommend that the authors clear up these and resubmit to other conferences.


**Time Spent Reviewing:**

5

---

> ### Author Response · Authors · 2021-08-10
> **Authors response**
>
> Thank you for your detailed review and efforts towards improving this contribution.
>
> Regarding stringent regularization, your intuition is true but we think that there's a confusion. Corollary 2.3.1 concerns the **adversarial setup** in which $||\text{argmin}_{\theta} L_T(\theta)||_2$ can grow proportional to $T$ (see line 435) forcing $\lambda \sim 1/T$ for a non-vacuous bound. Our argument is not that our bounds lift this limitation, but rather that in the **stochastic setting** we are not interested in being competitive to all competitor vectors but only to the true parameter. Hence the need for a specific analysis of the stochastic setting that lifts this strict condition on $\lambda$. The -constant- parameter $\theta^*$ that we call "true parameter" doesn't exist necessarily in the adversarial setting.
>
> For the dependence on $d^2$, this was discussed in the answer to reviewer 3, we believe this can be brought down to $d$ by using different confidence intervals and will include a discussion in the revised version. For the sentence in line 499, your remark is valid and we apologize for the poor wording. $S_T$ is a random variable and can only be bounded with high probability. Indeed, we show (see Eq. (12) and Lemma C.1), that with high probability $\sum_{s=1}^{t}\left(\left(\theta_{t-1}-\theta_{*}\right)^{\top} x_{t}\right)^{2} = O(\log(T)^2)$ therefore, with high probability $S_T$ is of order $\log(T)\times \log(\log(T)/\delta)$ (and not $\log(T)^{3/2}$, this was a typo). Consequently, with high probability, $S_T$ is second order. Moreover, we agree that the improvement for small losses is relevant and we will mention it in the revised version. Note however that this technique also requires prior knowledge of all bounds to tune regularization for optimal bounds. For instance, this is required to tune $\beta$ for the ONS algorithm. Thank you for the reference to the result in the book LS'20, we weren't aware of it. Evidently this methods implies the same bounds for the SLB, with only an additional second order term. However, this can not be transferred to the regression setting. We also show in Fig. 3 that the additional term for Ridge's bound can make a notable difference in the empirical performance. The contribution in the *applications section is advocating for the use of the forward algorithm whenever possible instead of ridge for its empirical stability and improved bounds, the SLB problem is only an example.
>
> We now address the questions regarding clarity. The explanation in p3:84-86 was not very clear. We meant to say that because there is a $1/\lambda_{r_T}(G_T)$ term, the bound is not uniform over features unless a lower bound on feature norms is given. It is not merely a problem of rank and more about the magnitude of eigenvalues. In line 119 Y is indeed Y$^\text{r}$. Lemma C.2 is indeed equivalent to Corollary 8 of [``Online-to-Confidence-Set Conversions and Application to Sparse Stochastic Bandits'' by Abbasi-Yadkori, Pal, Szepesvari], we will add a reference. In Fig. 2 it is online (instantaneous) regret not cumulative online regret, we will name it instantaneous regret and define it to be more clear.

---

> > ### Comment · Reviewer_FknY · 2021-08-30
> > **quick question**
> >
> > I just updated my review, but I just realized I don't fully understand how Corollary 3.3.1 implies the regret lower bound in the stochastic setting. The regret R^*_{T,[-Y,Y]} is about the optimal worst-case regret of algorithms when the response y_t can be set arbitrarily, right? If so, I don't know how we can relate this to the stochastic setting because R^*_{T,[-Y,Y]} involves sequences of $y_t$ that would receive very small probability mass in the stochastic setting. Please let me know if I am missing something.

---

> > > ### Author Response · Authors · 2021-08-31
> > > **Rectification Corollary 3.3.1**
> > >
> > > Thank you for your answer and for updating your score. There is a typo in Corollary 3.3.1: "for all $Y>0$" should be omitted, we apologize for this oversight. Indeed, in the stochastic setting $Y$ is the random variable which value at time $T$ is: $Y=\max_{1\le t\le T}|y_t|$ (see line 185). The notation $R_{T,[-Y,Y]}^\star$ is confusing in this setup, it is meant to represent inf_{ \mathcal{A} } sup_{ x_{1, ...,T } \in [0, 1]^d } sup_{(y_t)_{ 1 \le t \le T} } \{ \sum_{t=1}^T (y_t - \hat{y}_t^{\mathcal{A}})^2 - \inf_{ u \in \mathbb{R}^d } \sum_{t=1}^T (y_t - x_t^\top u)^2 \}, where $y_t$ is a $\sigma$-sub-Gaussian random variable of mean $x_t^\top \theta_*$. We will give it the different and simpler name $R_T^\star$. The proof of corollary 3.3.1 is effectively made in the stochastic case. To summarize, $R_T^\star$ concerns sub-Gaussian observations and the corollary is therefore a valid lower bound for the stochastic setting, thank you for catching that typo. Finally, we believe that the paper doesn't comprise technical errors, and that we can include the important discussions we had with reviewers efficiently. Indeed, the camera ready version allows for an additional page, we will also mention the less crucial discussions in the appendices.

---

> > > > ### Comment · Reviewer_FknY · 2021-08-31
> > > > **.**
> > > >
> > > > Okay, I understand the update.
> > > >
> > > > I am sure there will still be a way to show log^2(T) lower bound for $\bar R_T$. However, even with your updated definition below, I don't see how one can claim a lower bound for the stochastic case.
> > > >
> > > > $\inf_{ \mathcal{A} } \sup_{ x_{1, ...,T } \in [0, 1]^d } \sup_{(y_t)_{ 1 \le t \le T} }  \sum\_{t=1}^T (y\_t - \hat{y}\_t^{\mathcal{A}})^2$ $- \inf\_{ u \in \mathbb{R}^d } \sum\_{t=1}^T (y\_t - x\_t^\top u)^2 $
> > > >
> > > > This quantity is guarding against the worst possible values of $y_t$. Specifically, $y_t$ here is a sort of 'local' nonstochastic variable that are effective under $\sup$ (and under which it takes arbitrary values). So, it is still not clear what Theorem 2.3. can imply for the stochastic case.
> > > >
> > > > I would like to see a very specific chain of inequalities that show $R^*\_{T}$ is less than $\bar R_T$, so I can be convinced..

---

### Official Review · Reviewer_jxuz · 2021-07-13

**Rating:** 6
**Confidence:** 4

**Summary:**

This paper considers the classical online linear regression setting. It revisits the analysis of two important algorithms, ridge regression and the forward algorithm (also known as the Vovk Azoury Warmuth forecaster). First, the adversarial analysis is revisited to argue why and where this analysis falls short in covering the stochastic setting in a satisfactory manner. Then, the authors provide novel high-probability bounds for both algorithms, a lower bound, and an experiment showcasing the performance of both algorithms for several regularization parameters. Finally, the paper contains an application of the new analysis in stochastic linear bandits.

**Limitations And Societal Impact:**

The authors adequately addressed the limitations and potential negative societal impact of their work.

**Main Review:**

The new high-probability bounds reveal the interesting phenomenon that, contrary to the forward algorithm, for small sample sizes the ridge algorithm is sensitive to the tuning of the regularization. This behavior appears to be confirmed by the experiment in section 4. However, there are some thing unclear to me. First, in Figure 2, the y-label states that it is the regret, but it does not appear to increase over time, contrary to what I would expect based on the theoretical results. Second, for $\lambda = 0.001$ there is an unexplained bump in the regret of the forward algorithm, which appears to contradict the claims of robustness of the forward algorithm and I would like to know if the authors have an explanation for this bump.

From a technical point of view the main novelty appears to be the use of the confidence ellipsoid method of [1] to bound the difference between the true parameters and the parameters of the algorithm.

The writing is relatively clear in most places, but I do want to urge the authors to include a proof sketch of at least one of the high-probability bounds. While the limitations of the adversarial analysis are clearly written down, I can not guess from the main text alone how one would prove the high-probability bounds.

The applications of the new analysis to stochastic linear bandits is quite nice. I like the fact that the boundedness assumption can be dropped, which I imagine is helpful to practitioners as choosing an appropriate bound can be quite bothersome.

I find the significance of these results difficult to judge. On one hand, the paper only contains no new algorithm but only a new analysis of classical algorithms. On the other hand, the analysis reveals interesting behavior of these algorithms which I would not have predicted based on the existing adversarial analysis alone. Furthermore, the application reveals that there is also a practical relevance of the new analysis, which could also mean the new analysis could be relevant in other applications of the ridge and forward algorithm.

As a final remark, the results of this paper suggest that there is a $O(d)$ gap between the upper and lower bounds. In lines 247 to 252 it is argued that the additional factor $d$ is due to the time-uniformity of the upper bounds. Intuitively this does not makes sense to me and I want to ask the authors for some intuition behind this. Furthermore, the discussion in lines 247 to 252 seems to indicate that if we only want to provide results for some fixed $T$ the dependence on $d$ would improve from $d^2$ to $d$, is this true?

minor comments
- line 320: First --> first
- there are some unexpected bold symbols in Theorem 4.1 and equation (5)
- the Forward2ndTerm in Figure 1 has an unexpected drop off after 200 observations.
- the caption of Figure 1 should be improved as I do not think the plot shows the online regret.
- to improve the readability of the plots different line types would help


----- Post-rebuttal -----
The authors have adequately addressed some of my concerns. However, to address the concerns of the other reviewers I think that a major revision of this paper is required and therefore I will not raise my score.

**Time Spent Reviewing:**

5.5

---

> ### Author Response · Authors · 2021-08-10
> **Authors response**
>
> Thank you for your comments and remarks that help us improve the quality of this paper.
>
> In Fig. 2, what is plotted is the online (instantaneous) regret and not the cumulative regret as in the bounds. The latter decreases with time as the estimation of the parameter improves. Regarding the bump in Fig. 2b, it is only 10 times worse than the optimal, this is marginal compared to ridge regression where, for certain choices of $\lambda$ (see Fig. 2a), the instantaneous regret can be $10^6$ times larger than the optimal. We included the sketch of proof for Theorem 3.3. The latter is maybe not visible, we will detail it further in the revised version.
>
> Thank you for your remark on the uniform upper bound, our intuition about the extra $d$ factor was incorrect and we will include the following -correct- explanation instead. We thought that this factor was due to time uniformity because intuitively, from Corollary 2.3.1 for the forward algorithm, the leading term should scale with $d$ and not $d^2$. After we submitted the paper, we came across a refined result, namely Theorem 8 in [Lattimore, T., \& Szepesvari, C. (2017, April). The end of optimism? an asymptotic analysis of finite-armed linear bandits. In Artificial Intelligence and Statistics (pp. 728-737). PMLR.]. This theorem morally improves the confidence width in the estimation of the regression parameter from the $\sqrt{d\log(T)}$ of (Abbasi-Yadkori et al., 2011) to a term which is asymptotically $\sim \sqrt{\log(T)}$ thus removing the extra $d$. We will replace lines 247-252 by this factual explanation that shows the optimality of the dependence on $d$.
>
> Finally, thank you for the remarks regarding the writing and clarity, we will take them into account for the revised version. About the drop to zero of the Forward2ndTerm after 200 observations in Fig. 1, it is because in Eq. (5), the second sum's index stops at time $T-1$ while the first sum's stops at time $T$.

---

### Official Review · Reviewer_npqJ · 2021-07-15

**Rating:** 5
**Confidence:** 3

**Summary:**

This submission provides high-probability bounds for the Ridge and the forward algorithm in the setting of stochastic online linear regression. The advantage of the proposed bounds over previous results established for the adversary setting is that they do not scale with $Y$, the range of the observations. The authors further show that the bound of the forward algorithm has better scaling with the range of the features $X$ and the regularization coefficient $\\lambda$. The extension of the forward algorithm to stochastic linear bandit is also studied. The proposed algorithm enjoys better scaling on $X$ and $\lambda$ than the classical OFUL method with Ridge.

**Ethical Concerns:**

No ethical concerns.

**Limitations And Societal Impact:**

See the main review above.

**Main Review:**

==Novelty==
This paper provides new results for the online linear regression in the stochastic setting, which is free from the scaling of the observation $Y$. This advantage roots in using the self-normalized concentration to analyze the linear regression in the stochastic setting, which has been primarily used in a closely related topic: stochastic linear bandit (Abbasi-Yadkori et al., 2011). So, I do not find the results very surprising, and the technical contribution could be limited. It would be better if the authors could highlight the challenge to obtain the bounds or some new insights.

==Quality&Significance==
The theorems seem correct, but I do not check the proofs in detail. The theorems in the paper support authors' claim in most cases. But I still have some questions and concerns.

1) It seems the adversary bound of the forward algorithm (Corollary 2.3.1) implies an $O((\\log T)^2)$ bound without the scaling issue of $Y$ when the range of $x_t$ and $\\theta^*$ are bounded. One can quickly bound the $\\max{\vert y_t\vert}$ by $O(\\sqrt{\\log T})$ with high probability, since $\\vert y_t\\vert\\leq XS + \\vert\\epsilon_t\\vert$ and $\\max\\vert\epsilon_t\\vert$ grows in the rate of $O(\\sqrt{\\log T})$. So, the proposed bound might not have very significant advantage over the previous bounds in such a case.

2) I am unsure whether the scaling of the regularization coefficient $\\lambda$ matters in the SLB problem. Choosing $\\lambda$ as a constant seems good enough to obtain a $O(\\sqrt{dT}\\log T)$ bound for both Ridge and the forward algorithm. Meanwhile, the choice of $\\lambda = 1/T$ in the experiment is unfair for the Ridge algorithm since it would lead to a linear regret as suggested by the bound. So, I think it would be better to compare the algorithms with different $\\lambda$ (especially when setting $\\lambda$ as a constant).

==Clarify==
The paper is well-written and provides background on online linear regression in the adversary setting, preparing the reader well to understand their main arguments. But, I am not very sure is it good to use the word "unbounded rewards" for Theorem 4.1. The theorem still requires the upper bound of the range $X$ and the underlying models $S$. Combining the two upper bounds, one can show that Assumption 1 holds with certain scaling. So, the condition of Theorem 4.1 resembles the previous results in  (Abbasi-Yadkori et al., 2011).

Minor points:
line 306: "-as" -->"as"

The authors response has partially address my concern about the setting of $\lambda$. However, I still find that the paper needs a major revision to better discuss its motivation and contributions. So, I would like to keep my score.

====update post-rebuttal=====

The authors' response has partially addressed my concern about the setting of $\lambda$. However, I still find that the paper needs a major revision to discuss its motivation and contributions better. So, I would like to keep my score.

My original concern is that, for the stochastic setting, there is no need to use $\lambda = 1/T$ since a constant $\lambda$ is good enough to achieve the $O((\log T)^2)$ bound. In the rebuttal, I have just realized that the goal of this paper is to achieve the best-of-both-worlds for both stochastic and adversarial settings. The forward algorithm can achieve the goal with $\lambda = 1/T$ while the ridge algorithm would fail. I have to admit it is interesting to show the advantage of the forward algorithm in such a scenario. But, I think the author should provide a better discussion to highlight the motivation and contributions. In the current version, even the title "Stochastic Online Linear Regression: the Forward Algorithm to Replace Ridge" is somewhat misleading, since if we have known that the environments are stochastic, ridge algorithm with constant $\lambda$ seems good enough.



**Time Spent Reviewing:**

5-6 hours

---

> ### Author Response · Authors · 2021-08-10
> **Authors response**
>
> Thank you for your time and meaningful comments.
>
> Concerning the difference with similar results in the SLB setting. There the rewards are assumed to belong to $[-1,1]$ and this assumption is clearly required for the algorithm and the proof. This might make sense in for the SLB because actions are known therefore also an upper bound on their norms, but it doesn't make sense in the online regression scenario. Another difference is that in (Abbasi-Yadkori et al., 2011) the objective is the pseudo-regret, which dismisses the noise in observations. Moreover, Corollary 2.3.1 is not enough for ridge regression as it involves $Y^\text{r} = \max_{1\le t\le T} {|y_t|, |x_t^\top \theta_{t-1}|}$. The latter can only be bounded by $Y = \max_t |y_t|$ if it is known beforehand and predictions are clipped accordingly. In Theorem 3.2, $X$ is not known beforehand (*ie*. not required by the algorithm), it only appears in the bounds.
>
> The reviewer's intuition about the SLB is correct to some extent. A constant $\lambda$ is indeed the right choice, however, adversarial bounds advocate for a choice of $\lambda = 1/T$. The experiments that we perform here show that such a choice leads, as suggested by Theorem 3.1, to linear regret. The reason we conduct the SLB experiment with $\lambda=1/T$ is to demonstrate the robustness of the forward algorithm against the brittleness of Ridge. Indeed, as the reviewer suggested, for constant lambdas, both algorithms perform on par, but changing $\lambda$ leads to regrets of different magnitudes. Finally, we agree that using "unbounded rewards" for Theorem 4.1 in unclear. We clarify that it holds for unbounded rewards and will add the following explanation. First, we can drop the dependence on $X$ and $S$ by bounding the second term in the index of OFUL and OFUL$^\text{f}$ (see line 285) by $XS(1+X / \sqrt{\lambda})$ and then dropping this -constant- term from the index at the expense of a looser index. Therefore, knowing the bounds $X$ and $S$ is not crucial. Furthermore, while we choose to adopt the pseudo-regret definition like in (Abbasi-Yadkori et al., 2011), we could also derive similar bounds for the actual regret involving rewards $y_{t}=\left\langle x_{t}, \theta^{*}\right\rangle+\epsilon_{t}$ instead of their expected value, $(y_t)_{t\ge 1}$ are unbounded.

---

### Official Review · Reviewer_um9E · 2021-07-16

**Rating:** 5
**Confidence:** 3

**Summary:**

The paper offers a new analysis of the forward algorithm in the setting of stochastic online linear regression. Traditional analysis of this method under adversarial setting suffers from assumptions of boundedness of the output variable. These assumptions are not satisfied in the stochastic setting. The authors consider the case where there exists an unknown true parameter $\theta$ such that y = \theta^{T}x + \eta, where the noise term $\eta$ is sub-gaussian. They give explicit bounds on regret (in expectation and high probability bounds) and show its application in the specific problem of Linear Bandits.


**Ethical Concerns:**

None to my knowledge.

**Limitations And Societal Impact:**

The authors have done a great job in addressing the limitations and potential negative societal impact of their work.

**Main Review:**

The main contribution of the paper is its refined analysis of the Forward Algorithm for an alternate setting and in supporting the assertion through experiments. It is very clear to understand and does not claim anything superficial. The work seems to be original.
As such there are no new algorithmic techniques or proof techniques that catch my attention or that I find novel. The paper’s significance is hard to estimate, but at present, I find it to be a moderate increment over our understanding of the Forward Algorithm. I am of the opinion that this paper is good, but may not reach the acceptance level of this conference.   1. Apart from Linear Bandits, are there other possible use cases where FA trumps over Ridge?
2. Is it a correct assertion to make that for sub-gaussian noise, FA will always be better (as just as good) as Ridge?


After Rebuttal: I would express my gratitude to the authors for their clarifications. I have read the author rebuttal and the other reviews. I tend to keep my original score, since I think that this paper falls marginally short of this conference's acceptance threshold.


**Time Spent Reviewing:**

5 hours

---

> ### Author Response · Authors · 2021-08-10
> **Authors reponse**
>
> We thank you for your time and feedback.
>
> We believe the significance of this contribution stems from new evidence that the forward algorithm is robust to regularization and that its regret does not scale with the upper bound on observations. Just as important is the analysis of ridge regression that allows obtaining regret bounds without prior knowledge of a bound on observations unlike in the adversarial setting. Regarding applications of this algorithm, it can be invoked whenever linear function approximation is needed such as for linear Markov decision processes or for graph bandits.
>
> To answer your second question, the theory shows that in first order, FA is always better or at least as good as Ridge. Empirically, their performances are very similar except in situations where the features' range is large or when the regularization's weight is small in which FA trumps over Ridge.

---

### Decision · Program_Chairs · 2021-09-28

**Decision:**

Accept (Poster)

**Comment:**

The paper concerns online linear regression algorithm -- ridge regression and the Forward algorithm a.k.a. Vovk-Azoury-Warmuth forecaster -- in the stochastic setting. Both algorithms work in the adversarial setting but require the range of labels to be bounded (additionally, RR needs to know the range in order to clip its predictions). The authors consider a stochastic well-specified model $y = \langle \theta^*,x \rangle + \xi$, with the noise $\xi$ being sub-gaussian. They show that RR and Forward work in this setting (high-probability regret bound) even though labels are potentially unbounded, with a better scaling of the bound with respect to the regularization and the range of inputs. It is also shown that RR is more sensitive than Forward to the right choice of the regularization for small sample sizes. Applications of their results to stochastic linear bandits are given (better regret).

Overall, the reviewers liked the paper and found these results to be novel and interesting. However, they also pointed at two issues, which made most of them to evaluate the paper below the threshold bar:
1. Limited significance. The paper does not introduce any new algorithms, and the proof techniques applied to RR and Forward were not found to be novel or surprising. Two of the reviewers pointed out that due to sub-gaussianity of the noise, the labels are w.h.p. bounded in the range $O(\sqrt{\log T})$ (for bounded comparator), so the proposed bound might not have significant advantage over the existing one. Furthermore the upper bounds have $O(d^2)$ dependence on the dimension as opposed to $O(d)$ for the adversarial case (the authors claim, though, that they can get down to $O(d)$ by an improved analysis in the revision).
2. There were problems with presentation reported, and suggestions that the paper should be improved and more discussion should be added (two of the reviewers were confused about the motivation and the actual contribution, especially around the discussion on tuning the regularization).

This is why I think the paper is not yet ready to be publish. However, I encourage the authors to resubmit their paper, taking into account the reviewers' remarks, in particular: to improve the presentation, extend the discussion of the main results, discuss the missing references to past work, and provide clear motivation around the scaling of the regularization coefficient.


**Consistency Experiment:**

NeurIPS has a long history of experimentation. In 2014, NeurIPS ran an experiment in which 10% of submissions were reviewed by two independent committees to quantify the randomness in the review process. This year, we repeated a variant of this experiment to see how the quality of the review process has changed over time.  This paper was part of the experiment and was therefore assigned to two committees (consisting of reviewers, an Area Chair, and a Senior Area Chair) that reached independent decisions.  If both committees made the same recommendation, this recommendation was followed. If a single committee recommended acceptance, the paper was accepted (with the exception of a few cases in which the other committee identified what we considered a fatal flaw, e.g., an error in a key result).

This copy’s committee reached the following decision: **Reject**

The other committee assigned to the paper recommended **Accept (Spotlight)**.  You can find the other set of reviews, along with any follow up discussion with the authors here:
https://openreview.net/forum?id=rDdb26AQ0SO